# Dynamic magnetic crossover at the origin of the hidden-order in van der Waals antiferromagnet CrSBr

Sara A. López-Paz [1,2] ✉, Zurab Guguchia [3], Vladimir Y. Pomjakushin[4], Catherine Witteveen [1,2], Antonio Cervellino[5], Hubertus Luetkens [3], Nicola Casati[5], Alberto F. Morpurgo [1,6] & Fabian O. von Rohr [1] ✉

The van-der-Waals material CrSBr stands out as a promising two-dimensional magnet. Here, we report on its detailed magnetic and structural characteristics. We evidence that it undergoes a transition to an A-type antiferromagnetic state below $T_N \approx 140$ K with a pronounced two-dimensional character, preceded by ferromagnetic correlations within the monolayers. Furthermore, we unravel the low-temperature hidden-order within the long-range magnetically-ordered state. We find that it is associated to a slowing down of the magnetic fluctuations, accompanied by a continuous reorientation of the internal field. These take place upon cooling below $T_s \approx 100$ K, until a spin freezing process occurs at $T^* \approx 40$ K. We argue this complex behavior to reflect a crossover driven by the in-plane uniaxial anisotropy, which is ultimately caused by its mixed-anion character. Our findings reinforce CrSBr as an important candidate for devices in the emergent field of two-dimensional magnetic materials.

Two-dimensional (2D) van der Waals materials have been identified to be excellent platforms to host new collective quantum states[1,2]. They are widely considered as promising materials for future quantum technologies by enabling the next generation of electronic nanodevices[3,4]. In particular, intrinsic two-dimensional magnets are intensively studied as key components for the realization of spintronics[5–7]. The stabilization of 2D magnetic materials with a high critical temperature down to the monolayer limit remain a standing challenge[8–14]. In addition, for a wider application of magnetic monolayers in spin-based electronic devices, semiconducting materials with suitable band gap values and high carrier mobility are highly desirable.

$CrX_3$ trihalides a priori present suitable bandgap values of 1.2–1.8 eV[15,16]. However, the exploitation of their electrical properties is limited by their flat bands, and the resulting low carrier mobility[17,18]. This contrasts with the highly dispersive bands observed in semiconducting

transition metal dichalcogenides[19,20], with exceptional hole mobility values[1]. The combination of chalcogen and halogen anions thus is a promising route for the realization of large bandwidth magnetic semiconducting materials. Furthermore, in such mixed-anion compounds[21], the relative arrangement of the heavy halides allows for a specific modification of the magnetic interactions by a targeted control of the magnetic anisotropy. In this line, the antiferromagnetic (AFM), mixed-anion, van der Waals material CrSBr[22] stands out by combining a sizeable direct band gap of $\Delta E \approx 1.8$ eV with an exceptionally large bandwidth[23,24], and thus an expected high carrier mobility[25]. Furthermore, CrSBr exhibits a substantial air-stability and a high magnetic critical temperature of $T_N \approx 133$ K in bulk, predicted to be even higher in the monolayer[24–26]. A substantial magnetoresistance has been indeed demonstrated below the ordering temperature[27]. The potential application of CrSBr for spin-based electronic devices is further reinforced

[1]Department of Quantum Matter Physics, University of Geneva, CH-1211 Geneva, Switzerland. [2]Department of Chemistry, University of Zurich, CH-8057 Zurich, Switzerland. [3]Laboratory for Muon Spin Spectroscopy, Paul Scherrer Institut, CH-5232 Villigen PSI, Switzerland. [4]Laboratory for Neutron Scattering and Imaging, Paul Scherrer Institut, CH-5232 Villigen PSI, Switzerland. [5]Laboratory for Synchrotron Radiation - Condensed Matter, Paul Scherrer Institut, CH-5232 Villigen PSI, Switzerland. [6]Department of Applied Physics, University of Geneva, CH-1211 Geneva, Switzerland. ✉e-mail: sara.lopezpaz@unige.ch; fabian.vonrohr@unige.ch

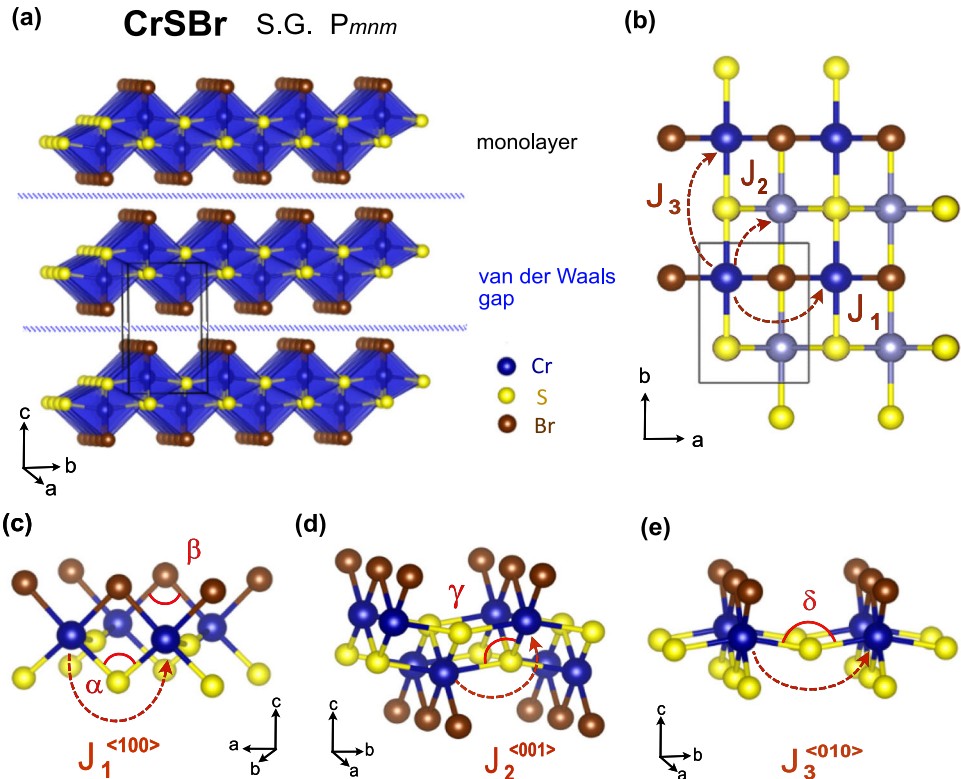

**Fig. 1 | Structure and Bonding in CrSBr. a** Crystal structure of CrSBr within the *Pmnm* space group (S.G.) showing the coordination polyhedra around the Cr atoms and the van der Waals gap separating the CrSBr monolayers. **b** Basal plane projection, showing the square-lattice arrangement of Cr cations within each layer. A lighter color is used for the Cr atoms on the lower layer. **c–e** The three main intralayer super-exchange paths (($J_1$), ($J_2$), and ($J_3$)) and their respective bond angles ($\alpha$, $\beta$, $\gamma$, $\delta$) are highlighted.

by the possibility of exerting magnetic control over the interlayer electronic coupling[23]. On the other hand, the exotic quasi one-dimensional transport properties and the anisotropic optical properties of CrSBr[23,28] emphasize the potential of mixed-anion chemistry to enlarge the functionalities of 2D van der Waals materials.

Concerning the magnetic properties, the magnetization measurements evidence that CrSBr undergoes an AFM transition below $T_N = 133$ K, together with a soft ferromagnetic behavior under high magnetic fields[27]. Hence, previously an A-type AFM structure, comprising ferromagnetic Cr-bilayers (here we refer to this as a "monolayer") that couple antiferromagnetically across the van der Waals gap, has been proposed[22]. Beyond that the temperature-dependent magnetic properties of CrSBr remain so far unresolved. In particular, recent magneto-electric transport measurements show a change on the sign of the magnetoresistance by lowering the temperature below 40 K[28,29], which goes along with the occurrence of an additional, subtle increase of the magnetization in CrSBr below that temperature[27]. This unusual change in the magnetoresistance, in the absence of a well-pronounced phase transition, suggests that a subtle change on the spin structure might occur at low temperatures as the origin for this hidden order[30]. The possibility of further complexity in the magneto-electric properties of CrSBr at low temperatures thus deserves further consideration.

Here, we address these open questions by a detailed characterization of the temperature-dependent magnetic and structural properties of CrSBr by combining neutron scattering, muon spin relaxation spectroscopy, synchrotron X-ray diffraction, and magnetization measurements. We show that the material adopts a long-range A-type magnetic structure below $T_N \approx 140$ K that persists for the whole temperature range in the magnetically ordered phase. On top of this, we identify a complex dynamic magnetism, with a slowing down of the magnetic fluctuations by lowering temperature below $T_s \approx 100$ K, leading eventually to a spin-freezing process at $T^* \approx 40$ K, which we

identify as the origin for a hidden order. We furthermore show that the spin-freezing is accompanied by an uncommon negative thermal expansion of the *a*-axis. The origin of this low temperature crossover is discussed, with special consideration of the role of the uniaxial magnetic anisotropy in the exotic dynamic behavior. These magnetic and structural properties, together with the sizeable band-gap and the large anisotropy within the layers are widening the potential application of CrSBr for spin-based electronic devices.

## Results

### Determination of the magnetic ground state of CrSBr

In Fig. 1 the crystal structure of CrSBr is depicted. The material crystallizes in the FeOCl structure-type in the space group *Pmnm*. The structure consists of monolayers of CrSBr, which are bonded through van der Waals interactions along the *c*-axis (Fig. 1a). The monolayers are built up of edge-sharing [$CrS_4Br_2$] octahedral units, with an underlying squared lattice arrangement of Cr(III) cations (Fig. 1b). The chemical bonding along the basal directions involves Cr-(S,Br)-Cr paths along the *a*-axis with a cation-anion-cation angle of $\alpha$ ($\beta$) ≈ 95 (90)° (Fig. 1c), whereas $\delta \approx 160°$ Cr-S-Cr paths along the *b*-axis (Fig. 1e). Within the CrSBr bilayers, Cr-S-Cr paths connect the two Cr layers with $\gamma \approx 96°$, as depicted in Fig. 1d.

To identify the structure of the magnetically ordered phase, we have performed temperature-dependent neutron powder diffraction (NPD) measurements. In Fig. 2, the obtained NPD patterns at $T = 160$ K and 1.8 K are depicted along with the respective Rietveld refinements (see Supplementary Tables 1 and 2). The refined structure for the normal state at $T = 160$ K is in excellent agreement with the previously reported orthorhombic structure with cell dimensions $a = 3.5066(1)$ Å, $b = 4.7485(1)$ Å, and $c = 7.9341(2)$ Å with the space group *Pmnm*[22]. Upon lowering of the temperature, strong magnetic reflections are observed, in accordance with the establishing of a long-range magnetic structure

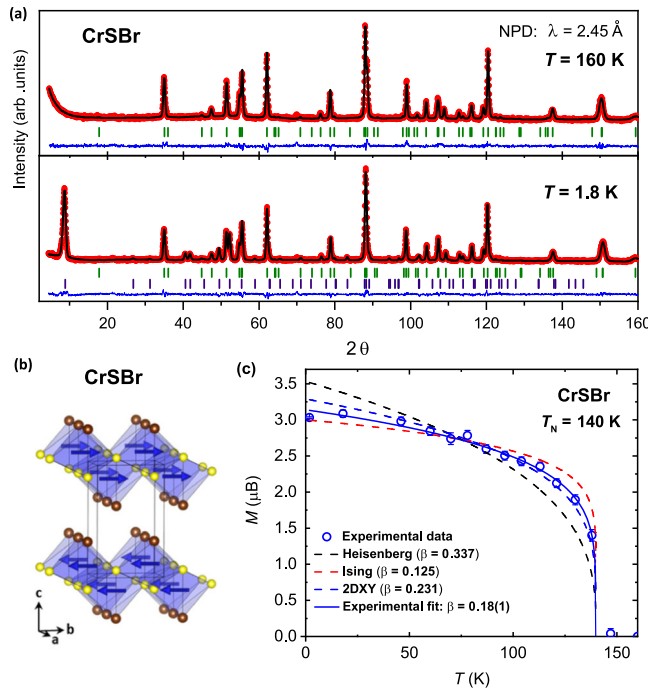

**Fig. 2 | Long-range magnetic order in CrSBr. a** Rietveld refinement of the neutron powder diffraction (NPD) data at 160 K (upper panel) and 1.8 K (lower panel) for CrSBr. The red dots correspond to the observed intensities, the black line to the calculated intensity and the blue line is the difference plot. Green and purple ticks show the Bragg reflections for the structural and magnetic phase, respectively. **b** Solved magnetic structure of CrSBr from NPD data. Sulfur and bromine atoms are shown in yellow and brown color, and Cr atoms are omitted for clarity. The magnetic moments are ferromagnetically aligned along the $b$-direction within the monolayer. The ferromagnetic layers are then interlayer coupled antiferromagnetically. **c** Temperature dependence of the refined magnetic moment (blue circles). The error bars represent the standard deviation of the fitted values. The dashed lines correspond to the power law $M \propto (T_N - T)^{\beta}$ with the different critical exponents of the corresponding models. The continuous blue line is the fit to the measured data.

with a propagation vector $\vec{k} = (0\ 0\ 1/2)$. This propagation vector corresponds to a doubling of the cell along the $c$-axis. From the Rietveld refinement of the 1.8 K NPD data, the magnetic structure is found to consist of an intralayer ferromagnetic alignment. Thereby, the magnetic moments are lying within the CrSBr monolayer along the $b$-axis. The interlayer interaction along the $c$-axis is on the other hand antiferromagnetic, forming an overall A-type antiferromagnetic structure, as shown in Fig. 2b. This is direct evidence of the previously suggested magnetic structure of CrSBr by an analysis of indirect magnetization measurements[22]. The refined magnetic moment of the NPD data at base temperature is found to be $M(1.8\ \text{K}) = 3.09(1)\ \mu_B$. This value is in good agreement with the one expected for the Cr(III) cations in octahedral environment, with $S = 3/2$ for a high spin state.

The temperature dependence of the refined magnetic moment is shown in Fig. 2c. It follows a power law behavior of the form $M \propto (T_N - T)^{\beta}$, allowing the analysis of the underlying order parameter in terms of the critical exponent $\beta$. In the case of magnetism with a pronounced 2D character, the Heisenberg model does not apply, but the Ising and the 2DXY universality classes are to be considered[31,32]. On one hand, systems with a pronounced preferential easy-axis direction (i.e., strong out of plane uniaxial anisotropy) fall into the Ising class with a predicted critical exponent $\beta = 0.125$. On the other hand, systems with easy-plane magnetic order are best described by the 2DXY class with a predicted critical exponent $\beta = 0.231$. In between, the crystal field can drive in-plane systems towards an Ising behavior,

resulting in intermediate values for the critical exponent $\beta$ between 0.125–0.231[32].

In the case of CrSBr, fitting of the magnetic moment derived from the temperature-dependent NPD data using this power law results in $\beta = 0.18(1)$ and a Néel temperature of $T_N = 139.8(6)$ K (see Supplementary Fig. 1). The Néel temperature is in excellent agreement with the magnetic susceptibility measurements (see Supplementary Fig. 2) and the critical exponent indicates a clear deviation from the Heisenberg behavior, as expected. This strongly points towards a two-dimensional magnetic character of the ordered phase. The obtained critical exponent indeed lies in between the 2DXY and Ising classes. Given the in-plane magnetization, this hints towards the presence of uniaxial anisotropy within the layers. The 2D character of the magnetic order in CrSBr in spite of the three-dimensional antiferromagnetic configuration is understood from the weak coupling across the van der Waals gap, as reflected in the low exchange coupling constants obtained by first principle calculations[24,33]. The weak interlayer coupling is also reflected in the occurrence of a low field meta-magnetic transition, first acting through decoupling of the ferromagnetic (FM) layers along the $c$-axis, as discussed below.

## Magnetic anisotropy and ferromagnetic correlations in CrSBr

The isothermal magnetization $M(H)$ curves for CrSBr at different temperatures are shown in Fig. 3a–c, for different relative orientations. A soft magnetic behavior is observed, with a complete field-induced polarization without hysteresis. At base temperature, the isothermal magnetization curves along the three different crystal axis (Fig. 3a) show a clear magnetic anisotropy. When the magnetic field is applied along the magnetic easy-axis (i.e., along the crystallographic $b$-axis) a sharp spin-flip transition is observed above $\mu_0 H_{\text{flip}} = 0.3$ T. On the other hand, when the magnetic field is applied along the other two main crystallographic directions a progressive decoupling is observed, with a linear increase on the magnetization. The substantial difference for the saturation field ($\mu_0 H^{\text{sat}}$) along the different crystal orientations further reflect the magnetic anisotropy within the in-plane directions, with anisotropy fields of 0.5 T, 1 T and 2 T for the three crystallographic $b$-, $a$-, and $c$-axis respectively. By increasing the temperature, the critical field for the meta-magnetic transition decreases (see Fig. 3b), as well as the saturation fields and the magnetization saturation values. The relative anisotropy fields along the three crystal-axes are on the other hand retained, with values of 0.3 T, 0.5 T and 1 T at $T = 100$ K along the $b$-, $a$-, and $c$-axis respectively. Finally, the magnetic anisotropy is lost by increasing the temperature above $T_N$, with an isotropic behavior for the three crystal-axes at 150 K as shown in Fig. 3c.

Considering the origin of the magnetic anisotropy, one first notices that the magnetic exchange paths along the $a$ and $b$ directions are clearly nonequivalent in CrSBr. In particular, a predominant contribution of the bromine atoms in the magnetic anisotropy through spin-orbit coupling (SOC) is expected[18,34]. For the related CrX$_3$ halides, an increase in the uniaxial anisotropy is indeed found when increasing the SOC effect, i.e., when going from the in-plane magnet CrCl$_3$ (0.02–0.03 meV; $T_c = 17$ K), to the out-of-plane CrBr$_3$ (0.11–0.19 meV; $T_c = 32$ K) and CrI$_3$ (0.68–0.80 meV; $T_c = 60$ K) counterparts[35–38]. A careful calculation of the magnetic anisotropy energy values for CrSBr show that the bromine contribution via SOC results in a clear uniaxial anisotropy[24], favoring the orientation of the spins along the $b$-axis, which is in agreement with the here determined magnetic structure from neutron data. On the other hand, the in-plane orientation – with an intermediate $a$-axis but a hard $c$-axis – results from a considerable shape anisotropy due to the layered character of CrSBr. The presence of uniaxial anisotropy in CrSBr differs from the model 2DXY behavior of the in-plane magnet CrCl$_3$[12], reinforcing the interest of CrSBr as potential host for exotic magnetic states.

Moreover, the high magnetization values and the S-shape observed in the $M(H)$ curves at $T > T_N$ (Fig. 3c) indicate that

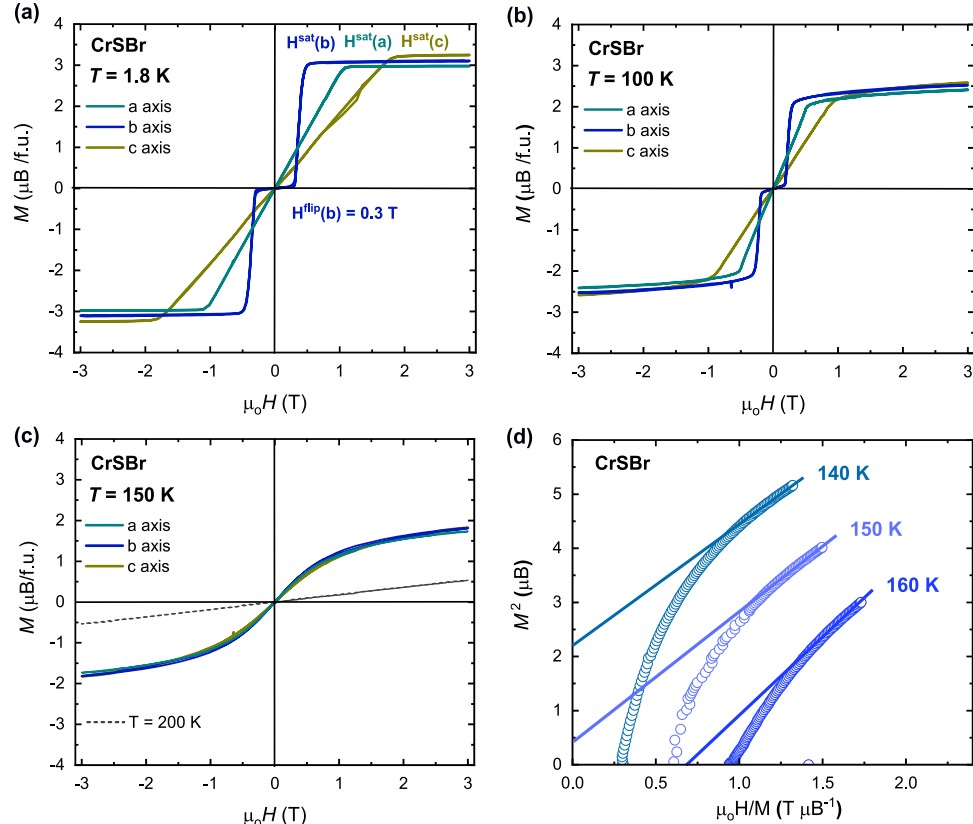

**Fig. 3 | Field-dependent magnetization of CrSBr.** Isothermal magnetization $M(H)$ curves for a CrSBr single crystal at different relative orientations are shown for different temperatures in **a**–**c** showing the absence of magnetic anisotropy at $T_N < T < T_M$. In **a**, the critical field for the spin-flip transition along the $b$-axis ($H^{flip}(b)$) and the saturation fields along the three main crystallographic axis ($H^{sat}(a, b, c)$) are shown. **d** Arrott plots for powder CrSBr at selected temperatures.

ferromagnetic correlations survive above the $T_N$. Hence, it is of substantial interest to analyze the soft ferromagnetic behavior under the applied magnetic field, as it allows considering the strength of the FM correlations within the layers. In fact, a higher magnetic critical temperature of $T_M > 150$ K is derived from the Arrott plots[39,40] shown in Fig. 3d. The lack of hysteresis, together with the loss of magnetic anisotropy inferred from the $M(H)$ curves above $T_N$, indicate this precursor magnetic state to reflect the inherent FM fluctuations within the CrSBr layers, without long-range coherence. The magnetization saturation values, $M(3T)$, can be fitted to a power law using a fixed exponent of $\beta = 0.18$ as obtained for the zero-field data (see Supplementary Fig. 3). An associated onset temperature $T_M = 153(6)$ K is obtained from this fitting. This onset temperature for the FM correlations agrees with the Arrott plots and is indeed in close agreement with the measured $T_c$ for isolated monolayers of CrSBr via second harmonic generation measurements[41].

Furthermore, we find an enhanced magnetic susceptibility above $T_M$. Free fitting of the magnetization saturation values to a power law (see Supplementary Fig. 4) gives indeed a higher critical temperature of $T_M \approx 175$ K, with critical exponent of $\beta = 0.29(8)$, now in between the 2DXY and the 3D Heisenberg model. Such a high critical temperature has been theoretically predicted for isolated CrSBr monolayers[24]. Our results clearly show that these high temperature magnetic correlations are already present in bulk CrSBr.

**Low-temperature uniaxial negative thermal expansion in CrSBr**
We turn now the attention to the low-temperature region, where the occurrence of a second subtle increase of the magnetization – below the main AFM transition – is inferred from the magnetic susceptibility measurements. In Fig. 4, we show the magnetic susceptibility for a CrSBr single crystal at different relative orientations, under a low

external magnetic field of 0.4 mT. A progressive increase in the susceptibility is observed along the three crystal axis by lowering temperature below $T^* \approx 40$ K. This magnetic transition occurs in the absence of a change in the average magnetic structure, i.e., there is no notable change in the intensity and position of the magnetic reflections in the NPD data.

Furthermore, this clearly distinct feature in the magnetization is not associated with any pronounced structural change, as there are no obvious changes to the structural reflections in the NPD data. For a high-precision evaluation of any subtle structural changes, we have furthermore performed temperature-dependent synchrotron X-ray diffraction (XRD) experiments. In Fig. 5a, the synchrotron XRD data at $T = 250$ K and 10 K as representative members are shown with the respective Rietveld refinements. All synchrotron XRD patterns (see Supplementary Fig. 5) are found to be in excellent agreement with the structure in the same $Pmnm$ space group. The temperature dependence of the obtained unit cell parameters (see Supplementary Table 3) is shown in Fig. 5b. The parameters are found to change gradually without any discontinuous change in the unit cell metrics, providing evidence for the absence of any distinct structural transition in the material. Interestingly, we observe an uncommon negative thermal expansion of the $a$-axis with a characteristic linear thermal expansion coefficient of $\alpha_a = -6.4 \cdot 10^{-6}$ K$^{-1}$. The increase in the $a$-axis by lowering temperature is followed by a comparable decrease in the $b$-axis with $\alpha_b = +10.9 \cdot 10^{-6}$ K$^{-1}$, until the in-plane cell parameters collapse at low temperature. The $c$-axis experiences a more pronounced shrinkage through the whole temperature range with $\alpha_c = +18.7 \cdot 10^{-6}$ K$^{-1}$, meaning a substantial reduction of the interlayer space as expected from the weak van der Waals interactions between monolayers.

It should be noted that a volume negative thermal expansion is observed in some prototypical 2D materials, i.e., graphene and

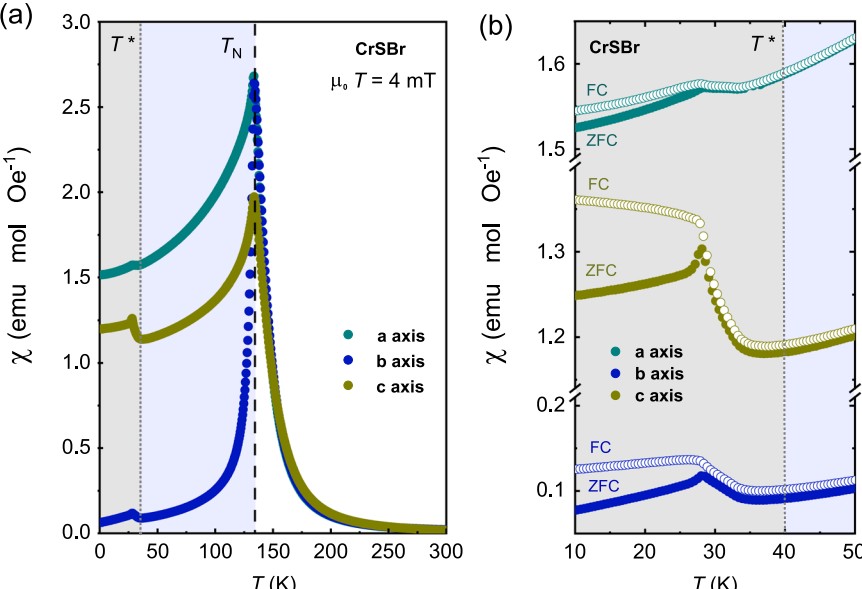

**Fig. 4 | Temperature-dependant magnetization of CrSBr. a** Magnetic suscept-ibility for a CrSBr single crystal at different relative orientations. **b** Enlarged low-temperature region showing the susceptibility in zero field cooling (ZFC; filled circles) and field cooling (FC; empty circles) modes. The magnetically ordered temperature regions are highlighted by blue ($T < T_N$) and gray ($T < T^*$) shadings.

CrBr$_3$[42,43], and that generally uniaxial negative thermal expansions are characteristic of highly anisotropic systems (i.e., chained cyanides and metal-organic frameworks)[44–47]. Here the uniaxial negative thermal expansion in CrSBr can be linked to the mixed-anion character of the material, with Cr-(S/Br)-Cr links along the $a$-axis. Hence, the mixed-anion chemistry is not only leading to a distinct magnetic anisotropy, as discussed above, but it is also resulting in a pronounced structural anisotropy. The presence of the more electronegative Br anions seems to also have a profound impact on the electronic properties of CrSBr, resulting in a different bonding character along the in-plane directions, i.e., with a more ionic character along the $a$-axis. The anisotropic band structure of CrSBr[25,41], with flat bands along the real space $a$ direction but dispersive bands along the $b$ direction, is then understood from an enhanced Cr-3d/S-3p hybridization along the $b$-axis as compared to the $a$-axis. The different transport properties observed along the two in-plane directions[28] further reflect the in-plane anisotropy in CrSBr, with high conductivity values along the $b$ direction while an insulating character along the $a$ axis. The highly anisotropic thermal expansion of CrSBr seems thus to emphasize its quasi one-dimensional character, going in line with the observed uniaxial magnetic anisotropy and the extremely anisotropic transport properties.

## Hidden order and spin-freezing in CrSBr

Using muon spin relaxation spectroscopy measurements, we have obtained a microscopic picture of the magnetic interactions in CrSBr. By following the time evolution of the muon spin polarization after implanting muons into the bulk of the crystal, the intrinsic magnetic response is obtained (see Supplementary Note 4). The ZF-$\mu$SR spectra shown in Fig. 6a–c display a spontaneous muon spin precession with a single frequency at low temperature, which is indicative for a long-range magnetically ordered state. The loss of initial asymmetry below the ordering temperature, as derived from the weak transverse field measurements, indicates a magnetic volume fraction of ≈90% below the $T_N(\mu SR) = 132$ K (see Supplementary Fig. 6). This observation is providing evidence of slow spin dynamics in CrSBr that reflect fluctuations of the Cr magnetic moments. The critical temperature obtained from the $\mu$SR data is lower than the $T_N(NPD) \approx 140$ K estimated from neutron diffraction. The strongly damped zero field (ZF) $\mu$SR spectra above 132 K (Fig. 6c) is however indicative of correlated

magnetic moments. This relaxation without oscillations might reflect fast dynamics which enter the $\mu$SR time-window (i.e., the MHz range) at a lower $T_N(\mu SR) = 132$ K. The exponential relaxation rate in the para-magnetic state, $\lambda_{pm}$, indeed shows a broad increase before it peaks at the $T_N$, as shown in Fig. 6d. The onset temperature for this precursor dynamic state is located at 160 K < $T$ < 180 K, in qualitative agreement with the magnetization measurements, where we have estimated a $T_M \approx 153$–175 K.

The internal field ($B_\mu$), as derived from the muon precession fre-quency ($B_\mu = \omega/\gamma_\mu$), can be considered as an order parameter in close relation with the internal magnetization. The temperature dependence of $B_\mu$, as derived from fitting of the ZF-$\mu$SR data, is shown in Fig. 6e. Below the $T_N$, the temperature dependence of the internal field can be approximated to a power law with a fixed model critical exponent of $\beta = 0.231$ (see Supplementary Fig. 7). By further lowering the tem-perature, an anomalous decrease in the internal field is clearly evi-denced below $T^* \approx 40$ K. It is worth stressing that the magnetic volume fraction remains unchanged below $T^* \approx 40$ K, and the ZF-$\mu$SR spectra are still well-fitted with the one oscillating component.

Further information about the temperature evolution of the internal magnetic field is obtained from the analysis of the temperature dependence of the oscillating fraction, shown in Fig. 6f. In particular, a continuous decrease in the oscillating fraction is observed below $T_s \approx 100$ K, with a weaker decrease below $T^* \approx 40$ K. This indicates that a continuous reorientation of the internal magnetic field occurs by low-ering temperature below $T_s \approx 100$ K, until it gets fixed below $T^* \approx 40$ K.

Complementary, the missing fraction as a function of tempera-ture is plotted in Fig. 6g, reflecting the additional loss of asymmetry at low temperature. The missing fraction is shown to experience a pro-minent increase below $T_s \approx 100$ K, before it saturates below $T^* \approx 40$ K reaching a ≈10% fraction. A phase separation between a ≈90% long-range ordered magnetic phase, and a ≈10% "disordered" magnetic state is then followed, the latter being responsible for the additional asymmetry loss.

The muon spin relaxation rates, $\lambda_1$ and $\lambda_2$, also show a complex temperature dependence within the ordered state, as shown in Fig. 6d. $\lambda_1$, which is the depolarization of the oscillating part of the spectrum, contains information mostly about the width of the static internal field distribution. On the other hand, $\lambda_2$ is associated to the

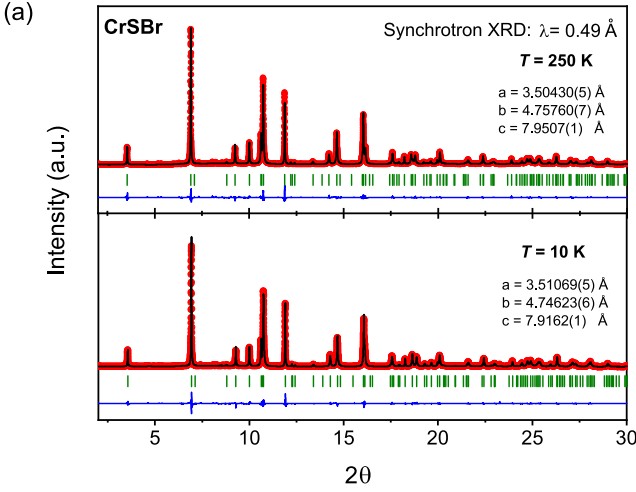

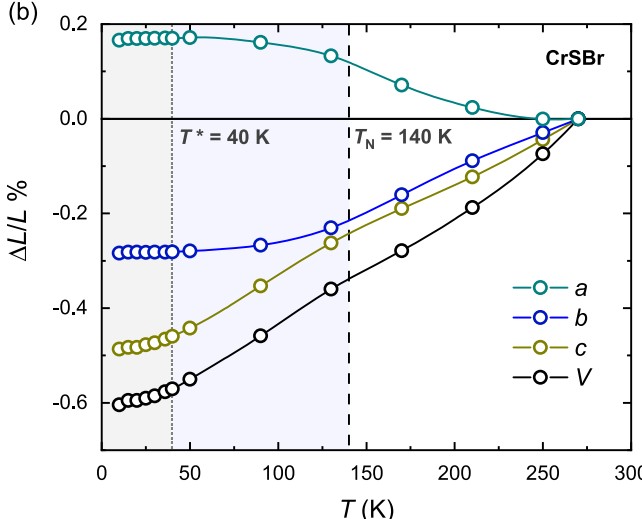

**Fig. 5 | Temperature-dependent structural data of CrSBr. a** The synchrotron X-ray diffraction (XRD) data of CrSBr measured with a wavelength of $\lambda = 0.49$ Å at $T = 250$ K (upper panel) and 10 K (lower panel), with the respective Rietveld refinements in the space group *Pmnm*. Red dots and black line correspond to the observed and calculated intensities, the blue line is the difference plot and green ticks show the Bragg reflections. **b** Normalized unit cell parameters *a*, *b*, and *c*, as well as the unit cell volume *V* in a temperature range between $T = 10$ K and 270 K. The magnetically ordered temperature regions are highlighted by blue ($T < T_N$) and gray ($T < T^*$) shadings.

non-oscillating relaxation, being therefore more affected by dynamic effects on the spin fluctuations. In the case of CrSBr, a progressive increase in $\lambda_1$ is observed below $T_s \approx 100$ K, indicating that the reorientation of the internal field is accompanied by a smooth increase in the width of the static internal field distribution. Concomitantly, $\lambda_2$ experiences a slight increase below $T_s \approx 100$ K, then followed by a steep rise below $T^* \approx 40$ K until a clear peak is observed at $\approx 20$ K. From the complex temperature dependence of $\lambda_2$ within the magnetically ordered state, a further slowing down of the magnetic fluctuations below $T_s \approx 100$ K can be derived, until a spin-freezing process occurs below 40 K. In this scenario, slow dynamics persist down to the lowest temperatures, i.e., in the quasi-static state below $T^* \approx 40$ K. The observed temperature dependence of $\lambda_2$ indeed agrees with an exemplary XY spin-freezing phenomenology[48], and the establishment of a quasi-static magnetic state at low temperature is consistent with the magnetic susceptibility measurements, showing clear hysteresis between the FC and ZFC measurements (see Fig. 4b).

We, however, notice that a second scenario, with a change on the static magnetic structure below $T^* \approx 40$ K, is to be also considered. Even though no change in the long-range magnetic structure is deduced from our NPD data, the phase separation deduced from the $\mu$SR data indicates that a change in the local magnetic structure might occur at low temperature. We therefore consider a combination of both scenarios to explain this low temperature hidden order in CrSBr, as discussed below.

## Discussion

The combination of the techniques applied here reveals that CrSBr displays a rich magnetic phase diagram as a function of temperature, which is comprehensively derived from the presented measurements and summarized in Fig. 7. The temperature-dependent NPD data reveals an average A-type AFM structure for CrSBr in the whole temperature range below $T < 140$ K (blue region in Fig. 7). Nonetheless, the dynamic character of the magnetic interactions is first reflected in the lower critical temperature derived from the $\mu$SR measurements (red dots in Fig. 7), according to the different time-windows of these techniques. A slowing down of magnetic fluctuations in the MHz-GHz range is thus deduced in the range $T_N(\mu SR) \approx 132$ K $< T < T_N(NPD) \approx 140$ K.

We understand this dynamic character to reflect a spin dimensionality (d) crossover in CrSBr. At high temperature (i.e., T > $T_N \approx$ 140 K), the fast dynamics reflect the presence of short-range ferromagnetic correlations between neighboring spins, framed on a Heisenberg picture where spin orientations along the three-crystallographic axes are possible ($d = 3$; see upper panel in Fig. 7). This precursor dynamic state may result in a fast damping of the observed muon signal, and justify the soft field-induced polarization without magnetic anisotropy. By lowering the temperature below $T_N \approx 140$ K, long-range coherence is established according to the determined A-type AFM structure. In this regime, fluctuations are still allowed but now confined to in-plane spin orientations with $d = 2$ according to a 2DXY model, as derived from the experimental critical exponent of $\beta \approx 0.18$. When these fluctuations slow down upon further temperature reduction to enter the $\mu$SR time window, oscillations are observed in the $\mu$SR spectra (i.e., at T < $T_N \approx 132$ K). However, the in-plane anisotropy may still favor the orientation of the spins along the crystallographic *b*-axis.

By further lowering the temperature below $T_s \approx 100$ K, our ZF-$\mu$SR experiments indeed show a continuous reorientation of the internal field down to base temperature, together with an increase in the ZF-$\mu$SR relaxation rates. We thus understand the spin reorientation to go along with a further slowing down of the fluctuations until a spin-freezing process takes place below $T^* \approx 40$ K. The concomitant decrease in the internal field, with a clear departure from the 2DXY model, might reflect a more defined uniaxial anisotropy, i.e., Ising like behavior in CrSBr at low temperatures with $d = 1$. The occurrence of this additional spin dimensionality crossover, giving a quasi-static directional spin structure at low temperature as depicted in Fig. 7, could explain the reduction of the internal field considering the time averaged moment to be different from the true static moment at low temperatures. The continuous change in the direction of the internal magnetic field until it gets fixed in the quasi-static state below $T^* \approx 40$ K further supports the proposed spin dimensionality crossover.

But, complementary, the possibility of a more complex magnetic state in the static region is also hinted by the observed phase separation to give an $\approx$10% "disordered" magnetic phase embedded in the long-range ordered predominant phase. In this line, it has been proposed that this low temperature hidden order in CrSBr, and the associated change in the magneto-electric properties, are associated to the magnetic ordering of electronic point defects at low temperature[29]. Nonetheless, the freezing into a frustrated magnetic state, as an alternative explanation for the low temperature magnetic state of CrSBr, might be also considered.

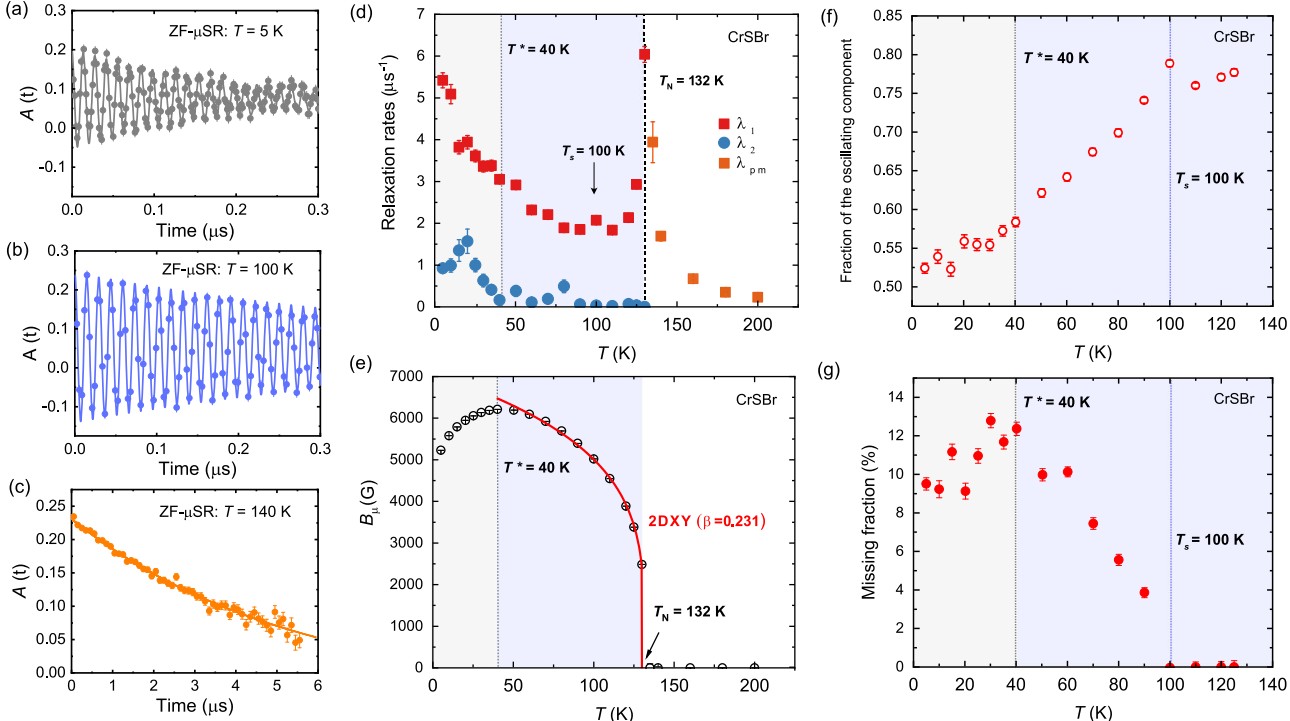

**Fig. 6 | ZF-μSR measurements on CrSBr. a–c** ZF-μSR spectra for CrSBr at different temperatures. Lines show fitting to Eqs. (3) and (4) (see Supplementary Note 4). Error bars in the μSR asymmetry are derived from the standard error of each bin over about ~$10^6$ events. **d** Temperature dependence of the muon spin relaxation rates $\lambda_{pm}$ (orange squares), $\lambda_1$ (red squares) and $\lambda_2$ (blue circles). **e** Temperature dependence of the internal field ($B_\mu$). Line shows fitting to a power law with a fixed critical exponent of $\beta = 0.231$ according to the 2DXY model. Temperature dependence of (**f**) the oscillation fraction and (**g**) the missing fraction. The magnetically ordered temperature regions are highlighted by blue ($T < T_N$) and gray ($T < T^*$) shadings. The error bars represent the standard deviation of the fitted parameters.

Our neutron diffraction data do not show any sign of a change on the spin structure, but the A-type structure to be retained down to base temperature. However, we cannot discard the formation of a frustrated magnetic state without long-range periodicity. Further investigation of diffuse scattering contribution by using polarized or small angle neutron scattering techniques will be necessary to address this possibility. The possibility of a frustrated magnetic state in CrSBr could be indeed anticipated by the analysis of the magnetic exchange interactions. First principle calculations predict positive exchange coupling constants for the first-neighbor interactions ($\langle J_1 \rangle$, $\langle J_2 \rangle$, and $\langle J_3 \rangle$) in Fig. 1)[24–26], in agreement with the experimental ferromagnetic configuration within the CrSBr monolayer. The ferromagnetic character of the Cr-Cr interactions along the a-axis follow the Goodenough–Kanamori–Anderson rules, given the involved Cr-S-Cr (Cr-Br-Cr) super-exchange paths with $\alpha(\beta) \approx 94(89)°$. The coupling within the monolayers along the c-axis, with $\gamma \approx 96°$, is also expected to be ferromagnetic from this argument. On the other hand, the super-exchange coupling along b-axis might result in competing FM and AFM interactions, as the Cr-S-Cr angle of $\delta \approx 160°$ significantly deviates from an ideal 180° angle. The presence of a competing AFM contribution along the b-axis could also explain the change from negative to positive magnetoresistance below ≈40 K observed in multilayers and monolayers of CrSBr[28,29].

Further studies in order to characterize the low-temperature magnetic ground-state of CrSBr are required. In particular, a further characterization of the spin dimensionality at low temperature by local magnetic probes such as nuclear magnetic resonance and neutron magnetic pair distribution technique would be of great interest to unravel the low temperature magnetic complexity. Complementary, understanding the ground state of CrSBr by means of Monte Carlo simulations accounting for the dynamic character of the magnetic interactions may provide further insights on the low temperature magnetic structure. This fundamental understanding, combined with the experimental characterization of the short range interactions will shed light into the driving force for the spin reorientation and eventual magnetic frustration of the A-type magnetic structure. Ultimately, understanding the role of the mixed-anion chemistry of CrSBr, with the resulting nonequivalent magnetic exchange paths, in the complex spin dynamics might allow exploration of further functionalities in low dimensional magnets.

In conclusion, we have characterized the temperature-dependent magnetic and structural properties of CrSBr by means of neutron scattering, muon spin relaxation spectroscopy, synchrotron X-ray diffraction, and magnetization measurements. CrSBr is shown to present a complex dynamic magnetic behavior, with a progressive slowing down of the spin fluctuations by lowering temperature. The main antiferromagnetic transition corresponds to the establishing of an A-type magnetic structure below $T_N$(NPD) ≈ 140 K, with a pronounced two-dimensional character as reflected by the low critical exponent of $\beta \approx 0.18$. Complementary, our μSR study clearly points out the occurrence of an additional low-temperature magnetic transition in CrSBr, with a critical slowing down of magnetic fluctuations below $T_s \approx 100$ K until a spin-freezing process takes place at $T^* \approx 40$ K. This hidden order is shown to happen within the average long-range A-type magnetic structure, suggesting a crossover towards a more uniaxial magnetic character at low temperature.

Overall, our findings reinforce that CrSBr is a promising van der Waals magnet with a strong uniaxial character in the magnetic, structural, as well as in the transport properties. This material may therefore open the door for exploring new applications, such as ultra-compact spintronics. On a broader scope, the inclusion of mixed-anion chemistry stands as a promising route for the design of new van der Waals materials with low dimensional magnetic character.

## Methods

*CrSBr bulk crystal growth.* CrSBr single crystals were grown by chemical vapor transport using elemental chromium (Alfa Aesar 99.99 %) and

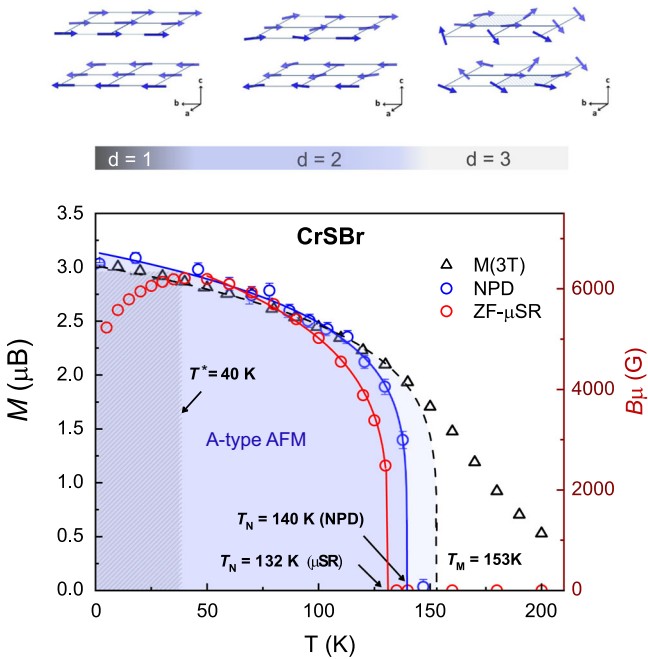

**Fig. 7 | Phase diagram of CrSBr.** Blue open circles show the magnetization values derived from the neutron powder diffraction (NPD) data, corresponding to the stablishing of the long-range A-type magnetic structure (blue region). The saturation values of the magnetization $M$(3T) are shown as black triangles, highlighting the precursor magnetic state below $T_M \approx 153$ K. Above $T_M$, high magnetization values are still observed, reflecting an enhanced magnetic susceptibility at high temperature (i.e., in the paramagnetic state). Red open circles show the internal field derived from the ZF-$\mu$SR data (right scale, in red), showing the decrease in the internal field ($B_\mu$) across the spin-freezing process. The error bars represent the standard deviation of the fitted parameters. Upper panel schematically show the proposed spin dimensionality (d) crossover.

freshly prepared $S_2Br_2$ in a 7:13 molar ratio, as reported elsewhere[49]. $S_2Br_2$ was prepared by reacting elemental sulfur (ACROS ORGANICS 99.999%) and bromine (ACROS ORGANICS 99+ %) under reflux in nitrogen atmosphere using a Schlenck line. The product was purified by vacuum distillation to remove unreacted bromine. The reactants were sealed under vacuum in a 20 cm long quartz ampule. After thermal treatment in a three-zone furnace with a temperature gradient of 950–880 °C for 140 h, CrSBr crystals were isolated at the middle-cold end of the tube. The needle-shaped black crystals were subsequently washed using warm pyridine, water and acetone. *Neutron diffraction experiments.* Neutron powder diffraction measurements were carried out using the high-resolution diffractometer HRPT at the Swiss Spallation Neutron Source (SINQ), Paul Scherrer Institute. The neutron wavelength of $\lambda = 2.449$ Å was used and the NPD data were analyzed using the Rietveld package FULLPROF SUITE and magnetic symmetry analysis using the BASIREPS software. The peak shape was modeled using a Thompson–Cox–Hastings pseudo-Voigt function with axial divergence asymmetry (as implemented in Fullprof; Npr = 7), using the instrumental resolution parameters characteristic of the diffractometer (https://www.psi.ch/en/sinq/hrpt/data-analysis). Due to layered character of the material, a preferential orientation was observed in the neutron diffraction patterns, that was refined using the modified March's function as implemented in Fullprof (Nor = 1). For the measurements single crystals of CrSBr were ground to fine powder (up to ≈ 1 g in weight) and placed in a vanadium can under ambient conditions. *SQUID magnetometry.* Magnetization curves and zero-field-cooled/field-cooled susceptibility measurements were carried out in a SQUID magnetometer (Quantum Design SQUID MPMS3) equipped with the vibrating sample magnetometer (VSM) option. The measurements

were performed in a temperature range between $T = 1.8$–300 K in sweep mode at a 2–5 K/min rate and 5–200 Oe/s. *Synchrotron X-ray diffraction experiments.* Temperature-dependent synchrotron XRD measurements were performed using the Materials Science (MS) X04SA beamline at the Swiss Light Source (SLS, PSI Switzerland). A Si NIST640C standard was used for precise determination of the wavelenght [$\lambda = 0.492355(5)$ Å] and for displacement and zero offset corrections, using the same experimental configuration as for the CrSBr capillary. The CrSBr powder sample was filled in a 0.3 mm capillary, and the experiments were carried out in the temperature range 10–270 K with a continuous rotation of the capillary. Diffraction patterns were collected upon heating from 10 K to 270 K, waiting 3 min for thermalization at each temperature before collection. Data was analyzed by the Rietveld method using the FULLPROF SUITE package. The profile parameters obtained from refinement of the Si NIST640C standard were used as a starting point for the peak shape modeling using the Thompson–Cox–Hastings pseudo-Voigt function. *$\mu$SR experiment and analysis.* Tranverse and zero field $\mu$SR experiments were carried out at the $\pi$M3 beam line (low background GPS instrument) of the Swiss Muon Source (SmuS) of the Paul Scherrer Insitute, using an intense beam ($p_\mu = 29$ MeV/c) of 100 % spin-polarized muons. Additional details can be found in Supplementary Note 4.

## Data availability
Data supporting the findings of this study are available within the manuscript and the Supplementary Information.

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

## Acknowledgements

This work was supported by the Swiss National Science Foundation under Grant no. PCEFP2_194183 F.O.v.R. A.F.M. gratefully acknowledges financial support from the SNF and for the EU Graphene Flagship project. The authors acknowledge helpful discussions with Harald O. Jeschke and Vanessa Kronenberg for valuable experimental help during the synthesis of this material.

## Author contributions

F.O.v.R and S.L. designed the experiments. S.L. and C.W. synthesized the crystals. S.L., Z.G., V.Y.P., A.C., H.L., N.C., and A.F.M. conducted the experiments. F.O.v.R and S.L. wrote the manuscript with contributions from all the authors.

## Competing interests

The authors declare no competing interests.
