## [Peer Review File · Nature Communications]

REVIEWER COMMENTS

Reviewer #1 (Remarks to the Author):

CrSBr is an van der Waals magnet of current great interest due to its high temperature magnetic order, a suitable semiconducting gap, and strongly anisotropic electro-magneto-optical properties. Such a material has promising applications in future quantum technologies and spintronic nanodevices.

This work first uses the NPD to characterize the crystal structure of CrSBr and to refine the A-type AF magnetic structure with the in-between 2DXY and Ising behavior. Then, using the magnetization measurements, the authors find the uniaxial magnetic anisotropy with the easy b-axis, intermediate a-axis, and hard c-axis, and the strong intralayer FM correlation. All these results are in good agreement with the available experiments and theories.

Moreover, using the XRD, the authors observe an interesting uniaxial negative thermal expansion along the a-axis at low temperatures, which is probably associated with the mixed S/Br character. Furthermore, based on the uSR, this work probes the fluctuation of the Cr magnetic moments and the slow spin dynamics, showing a spin freezing process from high-T 3D Heisenberg type, via the 2D XY, to the low-T 1D Ising one. The authors also suggest a possibly more complex magnetic state with a hidden order, which may arise from a magnetic frustration due to the competing FM and AF interactions along the b-axis.

This work presents a comprehensive study on the crystal structure of CrSBr and its rich low-dimensional magnetism. As such, I would recommend its publication in NC.

Just two minor comments:

1. On page 2, this sentence "Along the c-axis, the CrSBr monolayers are connected via ..." may be misunderstood as referring to the interlayer connection.

2. On page 5 bottom, the flat bands along the a-axis direction and the dispersive bands along the b-axis direction are most probably due to the much different bond angles, being about 90 degrees for the former (strongly bent) but 160 degrees for the latter (somewhat straight). The Cr-Br covalency could be stronger than the Cr-S covalency due to the fatter Br 4p orbitals.

Reviewer #2 (Remarks to the Author):

This is a well-written manuscript presenting a high-quality investigation into the structural and magnetic phase diagram of the van der Waals magnet CrSBr. A key noteworthy result is that the mixed anion nature of this material leads to distinct magnetic and structural anisotropies, the latter leading to a negative thermal expansion of the crystal structure. Thus, the interesting structural, magnetic and electronic properties of this material—combined with its two-dimensional nature—mean that this study will appeal to a broad readership across fields of materials science, chemistry and physics. I believe the paper is written well with this broad readership in mind, and the possible significance of this family of materials is well described.

This is an experimental study that is rigorous in that it brings together synchrotron X-ray diffraction, neutron diffraction, magnetometry and muon-spin spectroscopy measurements. The important results from these investigations are (i) confirmation of the crystal structure of this material, (ii) identification of its magnetic structure, (iii) confirmation of its magnetic anisotropy and dimensionality and (iv) discovery of persisting spin dynamics below the Neel temperature, pointing to complex magnetic behaviour at low temperatures and the possibility of a hidden order phase. These results are significant to the wider and emerging field of two-dimensional magnetism.

I believe the data collection and analysis have been performed to a high standard with sound methodology. However, there are a few points below on which it would be helpful for the authors to provide further insight or clarification before publication.

1. The power-law fits to extract critical exponents form an important part of the conclusions of this work, but it is not clear over which temperature range these fits have been applied in each case. Could the authors clarify the approach they have taken here and how sensitive their extracted fit parameters are to the region of data fitted?

2. In their discussion, the authors state, “Our neutron diffraction data do not show any sign of a change on the spin structure, but the A-type structure to be retained down to base temperature. However, we cannot discard the formation of a frustrated magnetic state without long-range periodicity.” Is there any evidence of any magnetic diffuse scattering in the neutron diffraction data?

3. When fitting the synchrotron X-ray data, was a standard used to extract lattice parameters as a function of temperature?

4. Also, regarding the Rietveld analysis of the powder diffraction data, how were peak profiles modelled? Did the authors experience any challenges in fitting the lineshapes due to the quasi-two-dimensional nature of the sample?

5. What further studies do the authors suggest as being useful to corroborate their hypothesis of spin dimensionality crossover described in the discussion?

Response to the reviewers' comments

Reviewer #1:

“CrSBr is an van der Waals magnet of current great interest due to its high temperature magnetic order, a suitable semiconducting gap, and strongly anisotropic electro-magneto-optical properties. Such a material has promising applications in future quantum technologies and spintronic nanodevices.

This work first uses the NPD to characterize the crystal structure of CrSBr and to refine the A-type AF magnetic structure with the in-between 2DXY and Ising behavior. Then, using the magnetization measurements, the authors find the uniaxial magnetic anisotropy with the easy b-axis, intermediate a-axis, and hard c-axis, and the strong intralayer FM correlation. All these results are in good agreement with the available experiments and theories.

Moreover, using the XRD, the authors observe an interesting uniaxial negative thermal expansion along the a-axis at low temperatures, which is probably associated with the mixed S/Br character. Furthermore, based on the uSR, this work probes the fluctuation of the Cr magnetic moments and the slow spin dynamics, showing a spin freezing process from high-T 3D Heisenberg type, via the 2D XY, to the low-T 1D Ising one. The authors also suggest a possibly more complex magnetic state with a hidden order, which may arise from a magnetic frustration due to the competing FM and AF interactions along the b-axis.

This work presents a comprehensive study on the crystal structure of CrSBr and its rich low-dimensional magnetism. As such, I would recommend its publication in NC. “

We thank the reviewer for his/her positive feedback and the valuable comments, as well as for recognizing the significance of our work to the broader scientific community.

Just two minor comments:

1) On page 2, this sentence “Along the c-axis, the CrSBr monolayers are connected via ...” may be misunderstood as referring to the interlayer connection.

We agree with the referee, and have adapted the sentence accordingly. **The description of the crystal structure in page 2 has been modified in the main text as follows:** “Within the CrSBr bilayers, Cr-S-Cr paths connect the two Cr layers with $\gamma \sim 96^\circ$, as depicted in Figure 1d”.

2) On page 5 bottom, the flat bands along the a-axis direction and the dispersive bands along the b-axis direction are most probably due to the much different bond angles, being about 90 degrees for the former (strongly bent) but 160 degrees for the latter (somewhat straight). The Cr-Br covalency could be stronger than the Cr-S covalency due to the fatter Br 4p orbitals.

We agree that structural distortions such as tilting or buckling of bonds may narrow the bands due to a reduced orbital overlap. However, the 90° connectivity along the a-axis is a usual configuration for face-sharing octahedral units, allowing in principle for extended covalent bonding. Despite its larger ionic radii, bromine presents also a higher Pauling's electronegativity than sulfur (i.e. 2.96 vs 2.58). Therefore a lower polarizability is expected (see, e.g. Kageyama, H. et al. *Nat. Commun.* (2018)). As a consequence, a more ionic character is expected for the Cr-Br bonds. From these considerations, we believe the ionic character of bromine is likely an explanation for the less dispersive bands along a-axis. **We have revised the main text, page 5, to clarify this aspect.**

Reviewer #2:

“This is a well-written manuscript presenting a high-quality investigation into the structural and magnetic phase diagram of the van der Waals magnet CrSBr. A key noteworthy result is that the mixed anion nature of this material leads to distinct magnetic and structural anisotropies, the latter leading to a negative thermal expansion of the crystal structure. Thus, the interesting structural, magnetic and electronic properties of this material—combined with its two-dimensional nature—mean that this study will appeal to a broad readership across fields of materials science, chemistry and physics. I believe the paper is written well with this broad readership in mind, and the possible significance of this family of materials is well described.

This is an experimental study that is rigorous in that it brings together synchrotron X-ray diffraction, neutron diffraction, magnetometry and muon-spin spectroscopy measurements. The important results from these investigations are (i) confirmation of the crystal structure of this material, (ii) identification of its magnetic structure, (iii) confirmation of its magnetic anisotropy and dimensionality and (iv) discovery of persisting spin dynamics below the Neel temperature, pointing to complex magnetic behaviour at low temperatures and the possibility of a hidden order phase. These results are significant to the wider and emerging field of two-dimensional magnetism.

I believe the data collection and analysis have been performed to a high standard with sound methodology. However, there are a few points below on which it would be helpful for the authors to provide further insight or clarification before publication.”

We thank the reviewer for his/her positive evaluation of our work and for the valuable comments.

1) The power-law fits to extract critical exponents form an important part of the conclusions of this work, but it is not clear over which temperature range these fits have been applied in each case. Could the authors clarify the approach they have taken here and how sensitive their extracted fit parameters are to the region of data fitted?

In some cases, the choice of the appropriate temperature window for the analysis of the critical behavior can have a significant influence in the determined critical exponents due to finite-size effects and dipolar interactions. In our case, the analysis of the neutron diffraction data was first approached by fitting the magnetization values to a power law in the 90-138 K temperature range, as an initial estimation of the T_N . From this analysis we obtain values of $T_N = 139.9(6)$ and $\beta = 0.18(1)$. We subsequently evaluated the validity of the critical behavior in the double logarithmic plot of the magnetization vs. the reduced temperature ($1-T/T_N$). A linear behavior is observed in the considered range for the fitting and holds for the whole temperature range, within the uncertainty of our measurements, supporting the applicability of the fitting. We have further confirmed that the determined critical exponent is robust against the selected temperature range for the fitting. In the case of the μ SR data, the observed dynamic contribution to the muon depolarization complicates a quantitative analysis of the critical behavior. Therefore, a qualitative analysis has been instead performed to estimate the onset temperature for the anomalous reduction of the internal field at low temperature. The internal field values in the 90-130 K temperature range were fitted to a power law with a fixed critical exponent of $\beta = 0.231$, assuming a model 2DXY behavior.

Additional information about the fitting procedure has been added to the main text (pages 2, 4 and 6) to clarify the considered approach in each case. Figures S1, S3-S4 and S7 in the Supporting Information now show the fitting window for clearness.

2) In their discussion, the authors state, “Our neutron diffraction data do not show any sign of a change on the spin structure, but the A-type structure to be retained down to base temperature. However, we cannot discard the formation of a frustrated magnetic state without long-range periodicity.” Is there any evidence of any magnetic diffuse scattering in the neutron diffraction data?

This is indeed an interesting point, we do not observe signs of diffuse scattering in our neutron powder diffraction data, and no evidence for satellite reflections or asymmetric peak broadening of the magnetic reflections are observed as a function of temperature. However, the background of our experimental data do not

allow to fully exclude this possibility. Considering that the frustrated magnetic region would account for a small volume fraction compared to the predominant long-range ordered phase, the possibility of capturing the associated diffuse scattering in our experimental configuration seems rather low. The use of polarized neutrons or a small angle neutron scattering configuration would be definitely more helpful in addressing the short-range order. **A sentence addressing this possibility has been added on page 9 in the main text.**

3) When fitting the synchrotron X-ray data, was a standard used to extract lattice parameters as a function of temperature?

For the calibration of experimental conditions in the synchrotron X-ray diffraction experiments a Si NIST640C standard was used. The standard was mounted on the cryostat, and a diffraction pattern was collected at room temperature and refined using the FullProf software to obtain the instrumental resolution parameters and for zero offset corrections and a precise determination of the wavelength [$\lambda = 0.492355(5) \text{ \AA}$].

The CrSBr capillary was then measured using the same experimental conditions. An initial diffraction data set was collected at room temperature and the sample was then cooled to base temperature, where an additional reference scan was collected. Diffraction patterns were then collected upon heating from 10 K to 270 K, waiting 3 min for thermalization at each temperature before collection. The sample temperature was monitored and the deviation within the set and readout temperature was found to be less than $\pm 0.1 \text{ K}$. **For better clarity, we have included this information in the methods section in the main manuscript.**

4) Also, regarding the Rietveld analysis of the powder diffraction data, how were peak profiles modelled? Did the authors experience any challenges in fitting the lineshapes due to the quasi-two-dimensional nature of the sample?

In the Rietveld refinement of the neutron powder diffraction data the peak shape was modelled using a Thompson-Cox-Hastings pseudo-Voigt function with axial divergence asymmetry (as implemented in Fullprof; $N_{pr} = 7$). The instrumental resolution parameters characteristic of the diffractometer (<https://www.psi.ch/en/sinq/hrpt/data-analysis>) were used as starting point for the refinement of the profile parameters. Due to layered character of the material, a preferential orientation was observed in the neutron diffraction patterns, that was refined using the modified March's function as implemented in Fullprof ($N_{or}=1$).

In the case of the synchrotron X-ray data, the profile parameters (half-width and Lorentzian contributions: U, V, W, X, Y) obtained from refinement of the Si NIST640C standard were used as a starting point for the peak shape modeling using the same Thompson-Cox-Hastings pseudo-Voigt function. In this case, the effect of the preferential orientation was minimized by a continuous rotation of the capillary.

We have included this information in the methods section in the main manuscript.

5) What further studies do the authors suggest as being useful to corroborate their hypothesis of spin dimensionality crossover described in the discussion?

We thank the Referee 2 for pointing out the interest of a further characterization of the spin dimensionality at low temperature. We believe that our results will spur work using several experimental magnetic probes, such as Nuclear Magnetic Resonance, local magnetic probe such as neutron magnetic pair distribution technique or magnetic imaging techniques such as scanning SQUID. Complementary, understanding the ground state of CrSBr by means of Monte Carlo simulations accounting for the dynamic character of the magnetic interactions may provide further insights on the low temperature magnetic structure. This fundamental understanding, combined with the experimental characterization of the short range interactions will shed light into the driving force for the spin reorientation and eventual magnetic frustration of the A-type magnetic structure. As this material is drawing substantial interest in the community, and there is great excitement for the manifold of properties exhibited by CrSBr, we are confident these still open questions will be soon addressed.

A sentence addressing these future directions has been included in the discussion, page 9 in the main text.

REVIEWERS' COMMENTS

Reviewer #1 (Remarks to the Author):

The authors have provided a satisfactory reply to both Reviewers comments, and have revised/improved their manuscript, accordingly. As this study offers an in-depth study on the crystal structure and the rich low-dimensional magnetism of the very interesting vdW material CrSBr, I would now recommend its publication in NC.

Reviewer #2 (Remarks to the Author):

The authors have fully addressed my initial comments in their response and the revised manuscript. As such, I now recommend the publication of this work.